# IMAGENETVC: Zero- and Few-Shot Visual Commonsense Evaluation on 1000 ImageNet Categories

**Heming Xia**[1,2][*] **Qingxiu Dong**[1,3][*] **Lei Li**[1,3]**, Jingjing Xu**[4]**,**
**Tianyu Liu**[5]**, Ziwei Qin**[1,3]**, Zhifang Sui**[1,3]

[1]National Key Laboratory for Multimedia Information Processing, Peking University
[2]School of Software & Microelectronics, Peking University
[3]School of Computer Science, Peking University    [4]Shanghai AI Lab    [5]Alibaba
{xiaheming,szf}@pku.edu.cn; {dqx,qinziwei}@stu.pku.edu.cn

## Abstract

Recently, Large Language Models (LLMs) have been serving as general-purpose interfaces, posing a significant demand for comprehensive visual knowledge. However, it remains unclear how well current LLMs and their visually augmented counterparts (VaLMs) can master visual commonsense knowledge. To investigate this, we propose IMAGENETVC, a human-annotated dataset specifically designed for zero- and few-shot visual commonsense evaluation across 1,000 ImageNet categories. Utilizing IMAGENETVC, we benchmark the fundamental visual commonsense knowledge of both unimodal LLMs and VaLMs. Furthermore, we analyze the factors affecting the visual commonsense knowledge of large-scale models, providing insights into the development of language models enriched with visual commonsense knowledge. Our code and dataset are available at https://github.com/hemingkx/ImageNetVC.

## 1 Introduction

With the breakthrough progress of Large Language Models (LLMs) in recent years (Brown et al., 2020; Zhang et al., 2022b), LLMs are gradually adopted as general-purpose API interfaces (e.g., ChatGPT[1]). In addition to language, these intelligent agents, are further required to understand vision knowledge (Hao et al., 2022), especially the visual perception, which is crucial for real-world interactions such as commonsense reasoning (Talmor et al., 2019), recipe generation (Agarwal et al., 2020), and robotic navigation (Shah et al., 2022).

However, current studies lack a systematic evaluation on how well these widely-used LLMs and their variants are capable of visual understanding. Recent research proposes to evaluate the visual capability of models through visual commonsense evaluation (Bagherinezhad et al., 2016;

---

[*]Co-first authors with equal contributions
[1]https://chat.openai.com

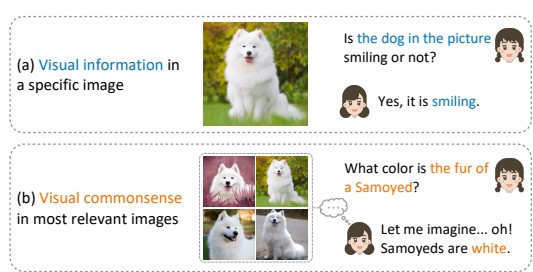

Figure 1: Illustration of Visual Commonsense. Visual commonsense refers to the general visual knowledge that is commonly shared across the world, as opposed to the visual information that is specific to a single image. Visual Commonsense can be captured through a series of related images.

Norlund et al., 2021). As shown in Figure 1, visual commonsense evaluation aims to evaluate the model's understanding of commonly shared human knowledge about generic visual concepts, including color (Bruni et al., 2012; Norlund et al., 2021; Zhang et al., 2022a), spatial relations (Liu et al., 2022), relative sizes (Bagherinezhad et al., 2016), etc. Despite their insightful investigations, these studies still have the following limitations from two sides: 1) **data side**: some research mines visual commonsense attributes based on frequency distributions in plain text corpora, which diverges from human visual perception and exhibits additional textual bias (Zhang et al., 2022a); 2) **model side**: most existing evaluations only focus on a specific model group, lacking a comprehensive exploration of various model families (Bagherinezhad et al., 2016; Norlund et al., 2021; Liu et al., 2022).

In this work, we propose that similar to human beings, models can also answer intricate visual commonsense questions with related images (illustrated in Figure 1). To this end, we introduce IMAGENETVC, a unified zero- and few-shot visual commonsense benchmark incorporating multiple sources of images (e.g., ImageNet (Deng et al., 2009), search images, and synthetic images). From

the data side, IMAGENETVC comprises 4,076 high-quality QA pairs, encompassing 1,000 ImageNet categories across diverse domains such as color, shape, material, component, etc. Moreover, as a human-annotated dataset, IMAGENETVC utilizes human visual perception to identify shared attributes across relevant images, avoiding textual bias and providing data that is more closely aligned with human knowledge. From the model side, besides unimodal LLMs, IMAGENETVC also enables the evaluation of various Visually-augmented Language Models (VaLMs) to investigate the effect of visual grounding, which compensates for the lack of images in previous benchmarks.

With IMAGENETVC, we conduct extensive evaluations on the most widely-used LLMs and VaLMs. We benchmark the visual commonsense capabilities of various LLMs such as OPT, LLaMA, and Falcon and assess the effect of visual grounding in VaLMs with multiple sources of relevant images. We further analyze the co-founding factors that may affect the visual commonsense capability of models, such as model scale, in-context learning, and image sources. We highlight several experimental findings. These findings support the high value of our benchmark in assessing visual commonsense capabilities.

- Template-based datasets yield artificially inflated and unstable visual commonsense evaluation, while our manually constructed IMA-GENETVC provides evidence that visual commonsense remains challenging for LLMs.
- We discover that the acquisition of visual commonsense is an emergent ability for LLMs. For instance, 1.3B could be a potential threshold for unimodal LLMs to emergent with visual commonsense on the component.
- In-context learning enhances the understanding of visual commonsense tasks for both LLMs and VaLMs, not only reducing their variance across prompts but also calibrating the model confidence on visual commonsense.

## 2 Related Work

**Large Language Models** Text-only Large language models (LLMs) have exhibited outstanding performance across various textual commonsense tasks, benefiting from their training on extensive textual data (Radford et al., 2019; Raffel et al., 2020; Brown et al., 2020). However, the lack of vi-

sual data (e.g., images) during pretraining restricts their visual commonsense capabilities (Li et al., 2023b). On the other hand, Visually-augmented Language Models (VaLMs) have gained popularity by integrating visual information into LLMs (Tsim-poukelli et al., 2021; Alayrac et al., 2022), which enhance the visual understanding capabilities of language models (Yang et al., 2022; Wang et al., 2022).

**Visual Commonsense Evaluation** Visual commonsense knowledge of visual concepts is a fundamental and critical aspect of AI systems seeking to comprehend and reason about the world (Yao et al., 2022; Dong et al., 2022). Previously, several datasets have been proposed to address specific attributes of visual commonsense, including MemoryColors (Norlund et al., 2021), ColorTerms (Bruni et al., 2012), RelativeSize (Bagherinezhad et al., 2016), and Spatial Commonsense (SpatialCS) (Liu et al., 2022). To evaluate general visual commonsense, Zhang et al. (2022a) introduced ViComTe, a template-based dataset consisting of various (subject, object) pairs (such as *(sky, blue)*). However, its reliance on pure textual input underestimates the visual capabilities of VaLMs. Furthermore, its utilization of template-based formats and automatic extraction techniques leads to substandard data quality and inherent textual biases.

In this work, we introduce IMAGENETVC, a human-annotated visual commonsense evaluation dataset that consists of 4K natural language QA pairs across various visual attributes, which supports both LLM and VaLM evaluation with multiple sources of images. We present detailed comparisons of IMAGENETVC with prior work in Table 1.

## 3 IMAGENETVC

Starting from ImageNet, we construct our IMA-GENETVC dataset in a multi-step crowd-sourcing pipeline, including 1) annotator training, 2) commonsense annotation, and 3) cross-check examination. An overall demonstration of our annotation process is illustrated in Figure 2.

### 3.1 Image Source

We selected ImageNet (Deng et al., 2009) as our image source because it covers a large number of commonly used objects in real-life situations, providing a diverse and representative image source. Additionally, the unified image format in ImageNet with

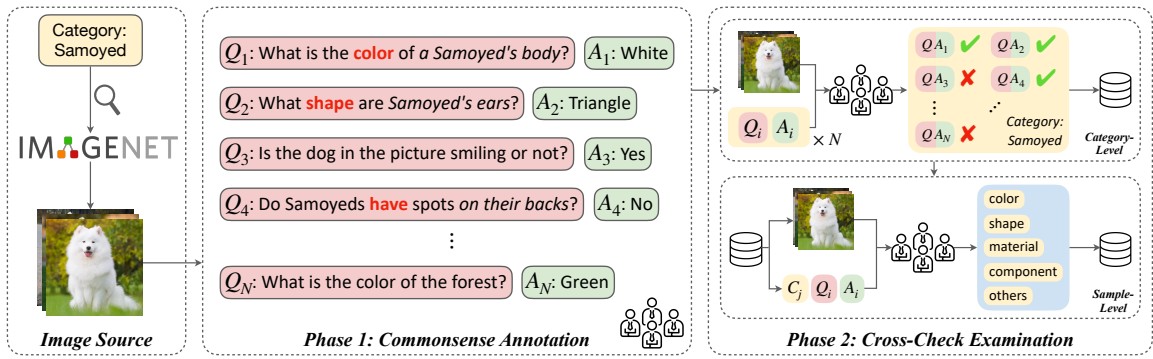

Figure 2: An overall demonstration of the construction procedures of IMAGENETVC.

| Dataset | Human Annotation | Multi-attribute Evaluation | Support VaLM | Region-based Question | Natural Language | #Category | #Test |
|---|---|---|---|---|---|---|---|
| MemoryColors (Norlund et al., 2021) | ✓ | ✗ | ✓ | ✗ | ✗ | 109 | 109 |
| ColorTerms (Bruni et al., 2012) | ✓ | ✗ | ✗ | ✗ | ✗ | 52 | 52 |
| RelativeSize (Bagherinezhad et al., 2016) | ✓ | ✗ | ✗ | ✗ | ✗ | 41 | 486 |
| SpatialCS (Liu et al., 2022) | ✓ | ✗ | ✗ | ✗ | ✗ | 59 | 1224 |
| ViComTe (Zhang et al., 2022a) | ✗ | ✓ | ✗ | ✗ | ✗ | 3957 | 2223 |
| IMAGENETVC (Ours) | ✓ | ✓ | ✓ | ✓ | ✓ | 1000 | 4076 |

Table 1: Features and statistical information of ImageNetVC and prior related datasets. The '# Category' column indicates the number of object categories included, and '# Test' means the number of test samples in the dataset.

dimensions of 256×256 pixels facilitates annotators' understanding of images and reduces feature engineering. Specifically, we used the widely-used ImageNet (ILSVRC) 2012 subset,[2] consisting of 1.4 million images from 1,000 object categories.

### 3.2 Prerequisite: Annotator Training

We posted online job listings on Amazon Mechanical Turk[3] and received over 500 applications from candidates with Bachelor's degrees or higher. To ensure dataset quality, we provided training with instructions and guidelines and a quick quiz to assess candidate understanding. Only candidates with scores larger than 95% are hired.

### 3.3 Phase 1: Commonsense Annotation

Figure 2 shows the commonsense annotation phase, where annotators are provided with category names and 50 randomly sampled images per category. They are instructed to form a question and answer considering shared visual features of the images and their own commonsense knowledge. Visual features may be object-based, such as the color of a entire object, or region-based, such as the color of a specific object part. Annotators first identify a common visual feature of the category, such as *The*

[2]image-net.org/challenges/LSVRC/2012/
[3]https://www.mturk.com/

*color of a Samoyed's body is white.* They then create a QA pair based on this feature if it aligns with their commonsense understanding of life, such as *What is the color of a Samoyed's body? White.*

To ensure that the QA pairs reflect visual commonsense rather than visual information tailored to specific images, annotators are instructed to focus on the visual features of each category rather than individual images. They are also provided with annotation examples and guidelines for rejection. The annotation UI and specifications for annotation can be found in Appendix A.

### 3.4 Phase 2: Cross-Check Examination

The primary objective of the cross-check examination phase is to conduct a rigorous screening and categorization of high-quality QA pairs that meet our requirements. This phase comprises two stages. In Stage 1, a category-level examination is performed, where three examiners are assigned to all annotated QA pairs in the same category. They are required to check all the pairs in the category based on the annotation instructions, rectify any grammatical errors, and eliminate low-quality or noncompliant pairs. Only the QA pairs that all three examiners approve are deemed acceptable. Stage 2 involves a sample-level examination. Although the examination in Stage 1 is efficient, examining

|  | Color | Shape | Mater. | Compo. | Others | Total |
|---|---|---|---|---|---|---|
| # Labels | 11 | 12 | 16 | 2 | 55 | 91 |
| # Categories | 439 | 367 | 405 | 560 | 768 | 1000 |
| # Samples | 557 | 424 | 430 | 1114 | 1551 | 4076 |

Table 2: Statistical information of IMAGENETVC. *Mater.* and *Compo.* are the abbreviations of Material and Component, respectively.

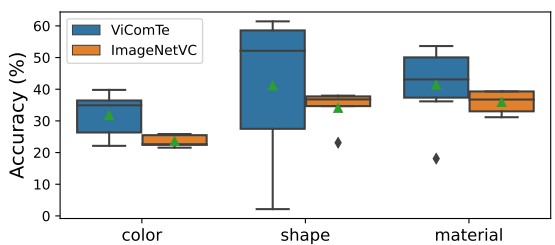

Figure 3: Zero-shot performance distribution of GPT-Neo-1.3B across different prompts. Green triangles represent mean results, black diamonds represent outliers. Compared to the template-based dataset, ViComTe, IMAGENETVC demonstrates notably reduced evaluation variance across various prompts.

all QAs in one category simultaneously creates a misalignment with the final testing method (one-by-one QA) and introduces a distribution bias in the examination process. Therefore, in Stage 2, a more thorough sample-level examination is carried out. Three examiners are randomly assigned a QA pair with the corresponding category name from the entire dataset. They vote on whether to accept the QA pair and classify it into the following five subsets: color, shape, material, component, and others. Only the sample that receives the majority vote is approved for acceptance.

Our 60-day annotated dataset comprises 4,076 items (refer to Table 2) from 1000 ImageNet categories. It consists of 5 individual sub-tasks: color, shape, material, component, and others. More information and examples of IMAGENETVC can be found in Appendix B. All pricing strategy details and a hierarchical supervision process employed are elaborated in Appendix A.3 and A.4.

### 3.5 Dataset Evaluation

Unlike previous datasets which are template-based, IMAGENETVC comes from diverse real images associated with human-annotated descriptions, which can better represent real-world settings. To assess the strengths of our dataset, we conduct automatic evaluation and human evaluation in this section.

First, we implement GPT-Neo-1.3B (Black et al.,

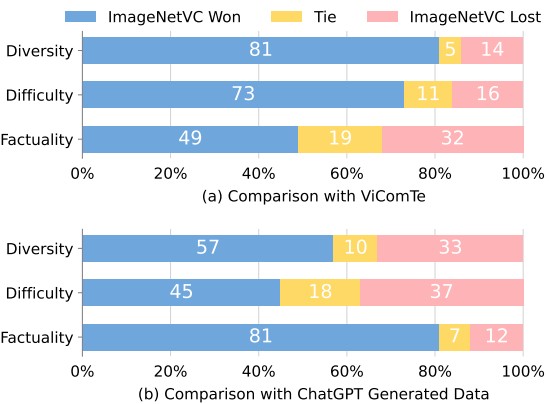

Figure 4: Human assessment of visual commonsense dataset from three aspects: diversity, difficulty, and factuality. IMAGENETVC outperforms ViComTe in terms of diversity and difficulty, while also demonstrating superior factuality compared to ChatGPT generated data.

2021) with respective subsets of IMAGENETVC and ViComTe, a widely-used dataset, across different prompts.[4] Results in Figure 3 indicate that, as a template-based dataset, ViComTe exhibits severe prompt bias, with substantial evaluation variance across different prompts. E.g., the model achieves only 2% accuracy with the prompt "X is of shape Y" but achieves 61% score with "The shape of the X is Y". Besides, compared with ViComTe, IMAGENETVC containing region-based questions is more challenging to models. For example, with the suitably selected prompt on the color subset, the model achieves 40% accuracy on ViComTe but only 28% accuracy on IMAGENETVC.

We further conducted a human assessment between ViComTe, IMAGENETVC, and QA pairs automatically generated from ChatGPT.[5] Specifically, we provided human annotators with sampled data from the two comparison datasets and asked them to vote for the better one considering diversity, difficulty, and factuality. As depicted in Figure 4, in more than 84% of cases, IMAGENETVC outperforms or matches the template-based ViComTe in terms of diversity and difficulty, which is consistent with the results in Figure 3. Moreover, our dataset demonstrates notably higher factual correctness compared to the data automatically generated by ChatGPT in more than 81% of cases.

To sum up, our data collection process ensured high-quality annotations with minimal bias and increased diversity, difficulty, and factuality, pro-

---

[4]Prompt details are provided in Appendix C.1.

[5]Please refer to Appendix C.2 for the detailed process of our designed human assessment.

viding a challenging dataset for advancing research in visual commonsense understanding.

## 4 Experiments

Our experiments primarily focus on two types of language models: LLMs and VaLMs. Both models have demonstrated promising capabilities in understanding visual information (Li et al., 2023b).

**Large Language Models** We begin with the text-only setting, to explore the visual commonsense capability of LLMs learned from the textual corpus. We focus on the dominant auto-regressive LLM family and benchmark a lot of model variants, including the GPT (Black et al., 2021; Wang and Komatsuzaki, 2021; Black et al., 2022), OPT (Zhang et al., 2022b), LLAMA (Touvron et al., 2023), Falcon (Almazrouei et al., 2023), and Pythia (Biderman et al., 2023).

**Visually-augmented Language Models** In our experiments, we mainly evaluate three widely-used open-source VaLMs: Z-LaVI (Yang et al., 2022), BLIP-2 (Li et al., 2023a), and MAGMA (Eichenberg et al., 2022). These VaLMs are mainly built on top of *frozen* LLMs and incorporate diverse mechanisms to integrate visual information. Further model details are provided in Appendix C.4.

### 4.1 Evaluation Methods

In this work, we focus on evaluating the zero- and few-shot visual commonsense of LLMs and VaLMs on IMAGENETVC. Following Schick and Schütze (2021) and Yang et al. (2022), we treat the zero-shot evaluation as a cloze test, transforming the QA pairs in IMAGENETVC into prompts like "[Question] The answer is [Answer]."[6]. Formally, each QA pair is converted into a sequence of tokens $\boldsymbol{x} = \{x_0, ..., x_i, ..., x_n\}$, in which $x_i$ is the answer. In the few-shot setting, examples with the same prompt are concatenated before each QA pair.

#### 4.1.1 LLM Evaluation

Given an LLM $\mathcal{M}$, the sequence of input tokens $\boldsymbol{x} = \{x_0, ..., x_n\}$ will first be mapped to text embeddings $\boldsymbol{e}_t = \{\boldsymbol{e}_t(x_0), ..., \boldsymbol{e}_t(x_i), ..., \boldsymbol{e}_t(x_n)\}$ by the embedding layer $\boldsymbol{e}_t \in \mathcal{M}$. Then we utilize the

---

[6]All the prompts utilized for the evaluation of LLMs and VaLMs are shown in Appendix C.1.

---

model to calculate the score for the answer $y \in \mathcal{Y}$:

$$
\begin{aligned}
s_t(y \mid \boldsymbol{x}) &= \frac{1}{-\log P_{\mathcal{M}}\left(x_i \mid \boldsymbol{x}_{<i}\right)} \\
&= \frac{1}{-\log P_{\mathcal{M}'}\left(\boldsymbol{e}_t(x_i) \mid \boldsymbol{e}_t(\boldsymbol{x}_{<i})\right)}
\end{aligned}
$$

where $\mathcal{M}'$ denotes the transformer neural network in $\mathcal{M}$, $P(\cdot)$ is the output probability given by the model. Then we obtain a probability distribution over all answer candidates using softmax:

$$
q_t(y \mid \boldsymbol{x}) = \frac{e^{s(y|\boldsymbol{x})}}{\sum_{y' \in \mathcal{Y}} e^{s(y'|\boldsymbol{x'})}} \tag{1}
$$

We calibrate the prediction by normalizing the probability distribution following Zhao et al. (2021), to mitigate the bias introduced by prompt formats as well as few-shot examples.

#### 4.1.2 VaLM Evaluation

We incorporate two types of image sources as additional visual inputs for evaluating VaLMs: images retrieved from the web and synthetic images. We adopt Google Image Search to retrieve relevant images and Stable Diffusion (Rombach et al., 2022) for image synthesis. Following Yang et al. (2022), for each QA pair, we utilize CLIP (Radford et al., 2021) to sort images from these two sources based on their similarity with the question and then preserve top-$K$ images as the final image sources. We mainly evaluate two types of VaLMs: *prefix-based VaLMs* and *ensemble-based VaLMs*.

**Prefix-based VaLMs** Given a QA pair with an image $\boldsymbol{v}$, *prefix-based VaLMs* (e.g., BLIP-2 and MAGMA) first utilize a visual encoder to transform the image into a sequence of visual embeddings $\boldsymbol{e}_v = \{\boldsymbol{e}_v^1, ..., \boldsymbol{e}_v^m\}$. Then, these embeddings are prefixed into the text embeddings of $\boldsymbol{x}$ and put into the *frozen* LLM backbone to calculate the score:

$$
\begin{aligned}
s(y \mid \boldsymbol{v}, \boldsymbol{x}) &= \frac{1}{-\log P_{\mathcal{M}}\left(x_i \mid \boldsymbol{v}, \boldsymbol{x}_{<i}\right)} \\
&= \frac{1}{-\log P_{\mathcal{M}'}\left(\boldsymbol{e}_t(x_i) \mid \boldsymbol{e}_v, \boldsymbol{e}_t(\boldsymbol{x}_{<i})\right)}
\end{aligned}
$$

The probability distribution with the image $\boldsymbol{v}$ is calculated over all answer candidates, which is the same as Eq (1). If $K$ images are provided, the final distribution will be averaged over all images:

$$
q(y \mid \boldsymbol{v}, \boldsymbol{x}) = \frac{1}{K} \sum_{i=1}^{K} q(y \mid \boldsymbol{v}^{(i)}, \boldsymbol{x}) \tag{2}
$$

Following Dong et al. (2023), for *prefix-based VaLMs* supporting few-shot evaluations, examples

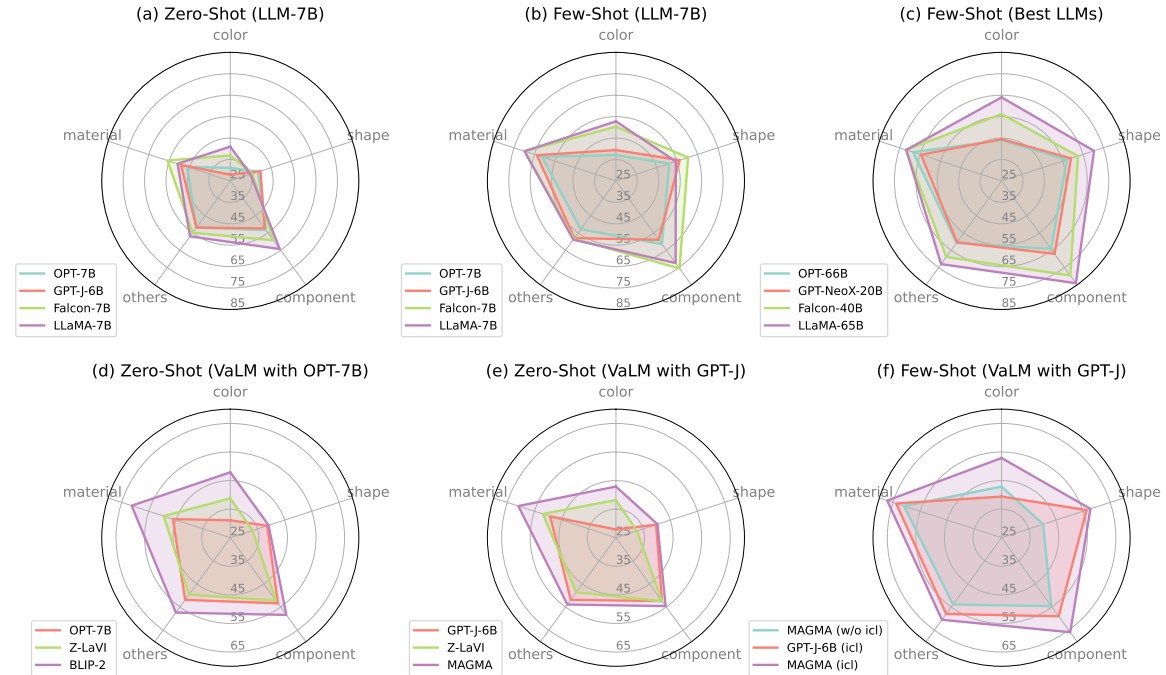

Figure 5: Radar plots for five individual sub-tasks in IMAGENETVC. We show evaluation results with four experimental settings: **(a, b)** Zero- and few-shot evaluation with LLMs-7B; **(c)** Few-shot evaluation with the best LLMs in their own model family; **(d, e)** Zero-shot evaluation with VaLMs and their *frozen* LLM backbones; **(f)** Few-shot evaluation with VaLMs. The numbers along the radio axis denote the mean Top-1 accuracy (%) of models over 5 different prompts. The detailed results for drawing these plots are shown in Appendix D.

$\{\boldsymbol{v}^{(j)}, \boldsymbol{x}^{(j)}\}_{j=1}^{L}$ with the same processing will be concatenated in front of each QA pair.

Since *prefix-based VaLMs* utilize *frozen* LLM backbones, they can be regarded as a conditional extension of text-only LLMs. Evaluations between these two model types facilitate a thorough assessment of the effect of visual grounding on visual commonsense ability.

**Ensemble-based VaLMs** Given the input tokens $\boldsymbol{x}$ and multiple images $\boldsymbol{v} = \{\boldsymbol{v}^{(i)}\}_{i=1}^{K}$, *ensemble-based VaLMs* (e.g., Z-LaVI) utilize a frozen CLIP model, which contains a text encoder $f_t$ and a visual encoder $f_v$, to project the tokens $\boldsymbol{x}$ and the image $\boldsymbol{v}^{(i)}$ into a shared representation space and compute the relevance score between them:

$$s_v(y \mid \boldsymbol{v}^{(i)}, \boldsymbol{x}) = \cos\left(f_t(\boldsymbol{x}), f_v(\boldsymbol{v}^{(i)})\right)$$

Then, same as Eq (1) and Eq (2), the probability distribution over all answer candidates and across $K$ images is obtained:

$$q_v(y \mid \boldsymbol{v}, \boldsymbol{x}) = \frac{1}{K} \sum_{i=1}^{K} \text{softmax}\left(s_v(y \mid \boldsymbol{v}^{(i)}, \boldsymbol{x})\right)$$

where softmax(·) is a simplified denotation of Eq (1). The final ensemble score is calculated as a weighted sum over the output distributions of LLMs and CLIP:

$$q(y \mid \boldsymbol{v}, \boldsymbol{x}) = (1-w) \cdot q_t(y \mid \boldsymbol{x}) + w \cdot q_v(y \mid \boldsymbol{v}, \boldsymbol{x})$$

where $w$ denotes the weight hyperparameter.

### 4.2 Experimental Details

We adopt Google Image Search to retrieve relevant images and utilize the newly released Stable Diffusion (Rombach et al., 2022) for image synthesis.[7] Following Yang et al. (2022), for each QA pair in IMAGENETVC, we obtain 100 images with each of the two methods. These 200 images are sorted using CLIP based on their similarity with the question. We preserve top-10 ($K = 10$) images for each QA pair as the final image sources. The other experimental details, such as the model implementation, hyperparameters, and computing resources are presented in Appendix C.3.

### 4.3 Main Results

The main evaluation results of LLMs and VaLMs on IMAGENETVC are shown in Figure 5. Here, we highlight several interesting findings.

---

[7] https://github.com/CompVis/stable-diffusion and we use the sd-v2-1 checkpoint.

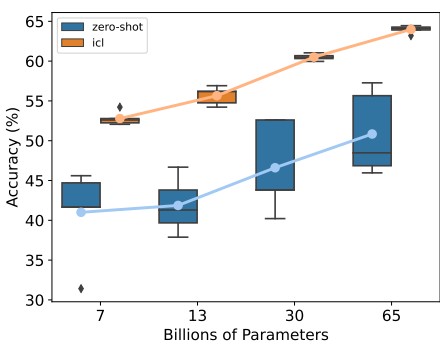

Figure 6: Performance distribution of LLaMA on the color subset. Models with ICL achieve higher performance and show reduced variance across prompts.

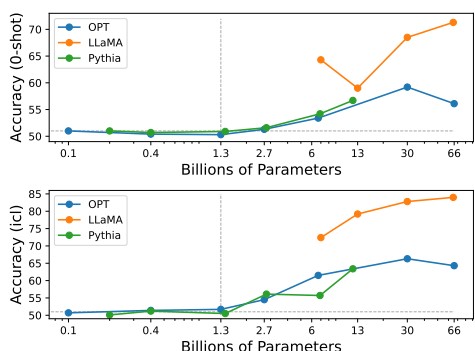

Figure 7: Performance of LLMs on the component subset of IMAGENETVC. LLMs with a model size larger than 1.3B demonstrate emergent capabilities when solving the component sub-task.

**Falcon and LLaMA excel in all four presented LLM model families, especially on the color and component sub-tasks.** As shown in Figure 5(a, b), Falcon and LLaMA consistently outperform OPT and GPT across various subsets with both experimental settings, despite the shape subset. Particularly, LLaMA achieves a zero-shot accuracy of 41% on the color subset, surpassing GPT-J with a considerable margin of 13%; Falcon yields the highest few-shot accuracy of 76% on the component subset, which amounts to 14% absolution improvement over OPT. We further present the few-shot results of the largest available LLMs in their own model family in Figure 5(c), where LLaMA-65B shows remarkable superiority over other counterparts.

**In-context learning (ICL) not only improves the visual commonsense performance of LLMs but also reduces their variance across different prompts.** Comparing the results in Figure 5(a) and 5(b), we found that given a few examples (i.e., with ICL), LLMs achieve consistent and remarkable improvement over themselves. For instance, LLaMA-7B with ICL achieves an average score of 62% across five sub-tasks, with a 12% improvement over the zero-shot result. We further show the performance distribution of LLaMA across different prompts in Figure 6, which illustrates that ICL not only improves the model's performance but also reduces its variance across different prompts. Further analysis is conducted in Section 5.

**VaLMs improve the visual commonsense ability of their LLM backbones, despite small performance gains on the shape subset.** As depicted in Figure 5(d, e), BLIP-2 shows remarkable superiority over OPT on the color and material subset, with average accuracy improvements of 17% and 15%, respectively, which indicates that incorporating visual information indeed helps to improve LLMs' visual commonsense capabilities. However, the results also show that the performance gains of VaLMs are small on some sub-tasks: both BLIP-2 and MAGMA only achieve an 0.5% accuracy improvement on the shape sub-task, while Z-LaVI even has performance drops. This demonstrates that VaLMs still have wide room for improvement.

**ICL capability of VaLMs should be further valued.** As shown in Figure 5(f), MAGMA with ICL achieves consistent improvements across all subsets over itself and the few-shot results of the *frozen* LLM backbone, indicating that ICL could also improve VaLMs' visual commonsense performances. However, ICL has been somewhat under-investigated by previous VaLM research. For example, both Z-LaVI and BLIP-2 only support zero-shot evaluation, with the lack of ICL capability. We hope our research could draw more attention to future work on the ICL capability of VaLMs.

## 5 Analysis

We further investigate the factors influencing the visual commonsense capabilities of LLMs and VaLMs. For instance, we find that a decent scale (e.g., 1.3B) could be a potential threshold for text-only LLMs to learn visual commonsense. We then analyze several influencing factors of VaLMs, such as image sources and image numbers.

**When (at what model scale) do text-only LLMs learn visual commonsense?** We show the zero- and few-shot results of three LLM families on the component subset in Figure 7. Take the component sub-task as an example, we find that a decent scale (e.g., 1.3B) could be a starting point for LLMs to emerge with visual commonsense on the com-

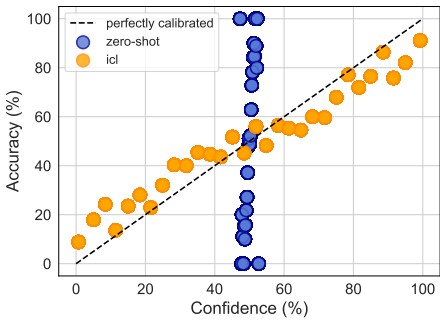

Figure 8: Calibration results of LLaMA-7B on the component subset. ICL greatly enhances model calibration, significantly boosting the correlation between confidence and accuracy from $r = 0.57$ to $r = 0.98$.

ponent[8]: smaller models at sizes below 1.3B are unable to perform well on the task, with a performance close to random guessing (i.e., ~50% accuracy); while models with a size larger than 1.3B exhibit gradual performance improvements. For example, OPT-30B achieves 59% and 66% average accuracy with zero- and few-shot settings, respectively.

**What is the effect of ICL on the calibration of LLMs?** Ensuring the reliability of a model's predictions is a crucial aspect of model evaluation, as mis-calibrated models can lead to incorrect inferences and serious consequences in real-world applications (Guo et al., 2017). To this end, we conducted a calibration analysis to evaluate whether the model confidence on visual commonsense reliably reflects the actual probability of the prediction being correct. Our analysis focuses on the calibration of LLaMA-7B on the component subset of IMAGENETVC. Results in Figure 8 indicate that ICL significantly improves the calibration of the model, increasing the correlation between confidence and accuracy from $r = 0.57$ to $r = 0.98$. This is consistent with the aforementioned findings, suggesting that ICL improves the visual commonsense performance of LLMs.

**How do image sources influence VaLMs?** Table 3 shows the ablation results of the image sources used in VaLMs. As illustrated, BLIP-2 with extra image sources (e.g., SEARCH) brings large improvements. This supports IMAGENETVC's motivation, suggesting that previous visual commonsense evaluations undervalue VaLMs' potential, as

---

[8] Please note that, as the evaluated LLMs (OPT and Pythia) both rely on the Pile (Gao et al., 2021) as their pre-training corpus, our findings may not be generalized to other LLMs.

|  | COL. | SHA. | MAT. | COM. | OTH. | AVG |
|---|---|---|---|---|---|---|
| OPT-2.7B | 25.8 | 39.9 | 40.2 | 51.3 | 48.1 | 41.1 |
| ✗ | 26.4 | 41.1 | 40.5 | 50.9 | 48.8 | 41.5 |
| RANDOM | 20.1 | 38.9 | 42.0 | 51.9 | 47.4 | 40.9 |
| SEARCH | 44.2 | 40.5 | 60.2 | 52.9 | 51.0 | 49.8 |
| SYNTHESIS | 43.2 | 39.5 | 59.2 | 53.8 | 51.3 | 49.4 |
| RANK | 44.7 | 40.3 | 61.9 | 54.0 | 51.7 | 50.5 |

Table 3: Zero-shot results of BLIP-2 with OPT-2.7B on IMAGENETVC with various image sources. ✗ means no image is provided to the model.

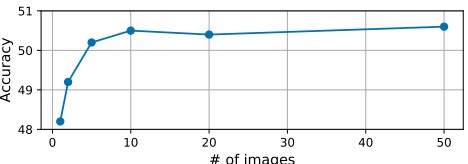

Figure 9: Average performance on IMAGENETVC with various numbers of top-ranked images. The results are obtained with BLIP-2 (OPT-2.7B).

VaLMs' visual commonsense demands relevant images as input to be suitably stimulated. Among the various image sources, CLIP-ranked images yield the best performance, suggesting that aligning images closely with the question facilitates the generalization of related visual commonsense by models. Thus, we use ranked images as the default image source for our main experiment and analysis.

**What is the typical number of images required to capture visual commonsense?** We show the model's average performance on IMAGENETVC with various numbers of images in Figure 9. The results show that the model's performance increases as the number of top-ranked images gradually increases from 1 to 10, indicating diverse image sources help the model to capture general visual commonsense. However, the improvement is marginal when the image number is larger than 10.

## 6 Visual Commonsense in Other Models

It is worth noting that, as a general visual commonsense dataset, IMAGENETVC supports various types of models and evaluation settings. Except for the evaluation setting in our main results, we also evaluate several models in the setting of open-ended generation. Specifically, we select two widely-used multimodal models, 12-in-1 (Lu et al., 2020) and BLIP (Li et al., 2022) that are finetuned on the VQAv2 dataset (Goyal et al., 2017) and the famous RLHF model, ChatGPT (gpt-3.5-turbo)

| Models | COL. | SHA. | MAT. | COM. | OTH. | AVG |
|---|---|---|---|---|---|---|
| 12-in-1 | 70.0 | 65.6 | 65.0 | 72.7 | 67.9 | 68.2 |
| BLIP | **77.0** | 70.0 | **73.9** | 71.5 | 69.7 | 72.4 |
| ChatGPT | 58.0 | 62.2 | 69.7 | 78.8 | 66.9 | 67.1 |
| ∟ w/ ICL | 67.9 | **75.0** | 72.3 | **87.5** | **76.2** | **75.8** |
| Humans | 91.2 | 92.7 | 90.5 | 98.9 | 94.0 | 93.5 |

Table 4: Evaluation results of multimodal models and ChatGPT on IMAGENETVC. We report Top-1 accuracy results obtained by open-ended generation. We also show human performance in the last row as a reference.

for evaluation.[9] As illustrated in Table 4, the multimodal models finetuned on VQAv2 show strong performance on IMAGENETVC, especially on the color sub-task, with relatively small model scales (e.g., 583M of BLIP). ChatGPT with ICL achieves the best average accuracy score of 75.8% across all compared models. However, it still has a considerable performance gap with humans, which has an average performance of 93.5%.

# 7 Conclusion

In this paper, we introduced IMAGENETVC, a comprehensive human-annotated dataset for evaluating visual commonsense using both textual and visual inputs. We conducted extensive experiments to evaluate the visual commonsense of both unimodal LLMs and VaLMs using IMAGENETVC. Our results demonstrate the varying degrees of visual commonsense knowledge present in different models, as well as the factors that contribute to the acquisition and enhancement of this knowledge. Additionally, we offer insights into the emergent abilities of LLMs and the strengths of VaLMs in the realm of visual commonsense.

# Limitations

While our study provides valuable resources and insights into the visual commonsense knowledge of LLMs and VaLMs, several limitations need to be acknowledged. Firstly, due to the high cost of region-based human annotation, the IMAGENETVC dataset only covers 1,000 ImageNet categories, which may not cover all real-world scenarios. Therefore, it is possible that models may perform differently on other types of images that are outside of the IMAGENETVC categories.

Additionally, our study is limited to zero- and few-shot visual commonsense evaluation and only considers models that have been pretrained on

---

[9]The evaluation details are illustrated in Appendix E.

large amounts of text data. Thus, it remains unclear whether fine-tuning on visual commonsense tasks would improve the performance of LLMs and VaLMs. Therefore, this may not fully capture a model's ability to understand visual commonsense in real-world scenarios where prior knowledge may be available. Besides, although we explored the factors that affect the visual commonsense knowledge of large models, it is still challenging to interpret how the models acquire this knowledge.

Overall, our study provides a foundation for future work in the field of visual commonsense knowledge and its applications. However, additional research is necessary to address the aforementioned limitations and further advance the field.

# Acknowledgements

This paper is supported by the National Key Research and Development Program of China 2020AAA0106700 and NSFC project U19A2065.

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

# Appendix

## A Annotation Details

In this section, we will provide a comprehensive overview of our annotation process, including the guidelines we follow, the user interface we use, the hierarchical supervision process we employ to ensure data quality, and our payment policy.

### A.1 Annotation Guidelines

The annotation of IMAGENETVC involves observing 20-50 images of a given category, finding a vision feature of the category, checking if it conforms to most of the images and our commonsense of life, and then writing a simple question-answer (QA) about this vision feature. The QA should contain one question and one correct answer.

The vision features can be object-based (such as the color, shape, material, and spotted/striped patterns of the whole object) or region-based (such as the color, shape, and material of a certain part of the object). They are features that can be seen through the images.

The annotation pipeline involves looking at the 20-50 images given and finding a common vision feature of the category. For example, "The shape of the dorsal fin of the tiger shark is triangle". The annotators check if this feature conforms to their commonsense of life and if it is written. Then, one QA is created, such as "What is the shape of the dorsal fin of the tiger shark? Triangle".

The following rules must be followed during the annotation process:

- The question should contain the name of the category. Otherwise, the submission will not be passed.
- If the annotator cannot think of a question that can be written or the images cannot be displayed, the annotator can skip this category.
- The first letter of the question needs to be capitalized.
- The end of the question needs to be a question mark.
- Please do not write lots of Yes/No questions. These questions are more likely to be rejected. We encourage the annotators to write more diverse answers.

In the annotation examples, we describe how the correct QA is created. Annotators can write their own QA according to this pipeline. The rejected examples include cases where the QA has been written before, vision features cannot be found in the images, or the QA is not about vision features. Additionally, the QA should conform to our commonsense of life, be strongly related to the category, and be about a specific feature of the category.

In summary, the annotation guidelines of IMAGENETVC involve observing images of a category, finding a vision feature, and creating a QA about the feature that conforms to our commonsense of life and follows the rules outlined above. These guidelines ensure the accuracy and consistency of the annotations, making the dataset suitable for use in various applications

### A.2 Annotation UI

Figure 11 shows the annotation user interface used in our human annotation process for model knowledge assessment. The interface consists of three main parts: the task instruction, the annotation pipeline, and the most common cases we reject. The task instruction provides clear guidance for the annotators on how to write effective prompts to assess the model knowledge. The annotation pipeline displays the generated text by the model and allows the annotators to refine their prompts until the generated text matches the expected target answer. The rejected cases section provides examples of prompts that do not meet the criteria and serves as a reference for the annotators to avoid such mistakes. The user interface design is intuitive and user-friendly, which greatly improves the efficiency and accuracy of the human annotation process.

### A.3 Hierarchical Supervision

To ensure high-quality annotation, we have implemented a hierarchical supervision process. During the annotation phase, examiners cross-check the annotation results, and annotators who are excessively rejected receive a warning. Those who exceed the warning limit are removed. Additionally, during the examination phase, a random sample check of the examination results is performed by five authors, and examiners with low-quality checks also receive a warning. The hierarchical supervision process guarantees the high-quality execution of the entire annotation process.

### A.4 Payment Policy

We compensated the crowd workers with varying rates based on the workload and quality of their

| Subset | Answer Candidates |
|---|---|
| Color | brown (tan), black, white, yellow (gold, golden), green, gray (grey), red, orange, blue, silver, pink |
| Shape | round (circle), rectangle (rectangular), triangle (triangular), square, oval, curved, cylinder, straight cone, curly, heart, star |
| Material | metal (steel, iron), wood (wooden), plastic, cotton (yarn, wool), glass, fabric (nylon, silk, rope) stone (rock), rubber, ceramic (porcelain), cloth (denim), leather, flour (dough, bread), paper, clay, wax concrete |
| Component | yes, no |
| Others (yes/no) | yes, no |
| Others (numbers) | 2 (two), 4 (four), 6 (six), 1 (one), 8 (eight), 3 (three), 5 (five) |
| Others (others) | long, small, short, large, forest (jungle, woods), water, ocean, big, tree (branch), ground, tall, wild outside, thin, head, thick, circle, brown, soft, land, neck, rough, chest, smooth, fur, hard, top, plants black, metal, books, vertical, lake (pond), grass, road, sky, front, kitchen, feathers, stripes, baby, hair feet, mouth, female, table |

Table 5: The answer set of all subsets in IMAGENETVC. Inside the parentheses are attributes grouped into the same answer candidate.

work. In Phase 1, workers were tasked with summarizing shared visual characteristics from 50 images and creating QA pairs. If their work was accepted in Phase 2, they would receive $0.50 for each sample. However, if their work was rejected, they would only earn $0.10 per sample. In Phase 2, annotators were paid $0.30 per sample for cross-checking tasks. On average, the annotators received approximately double the local minimum wage per hour.

## B  Details of IMAGENETVC

### B.1  Details of the Others subset

Annotated QA pairs that do not belong to the four specified sub-tasks (e.g., color, shape, material, and component) will be categorized into the Others subset. Therefore, The Others subset contains a more diverse range of QA samples, which is more challenging. Figure 10 illustrates the detailed composition of QA types in the Others subset, which covers various topics such as length comparison (21%), relative size (20%), living environment (16%), counting (12%), etc.

### B.2  Answer Set

Considering that the results of open-ended generation are uncontrollable, we evaluate all models with constrained decoding in our main experiments. Table 5 shows the list of all possible answers in IMAGENETVC. Besides, we noticed that LLMs tend to predict "yes/no" or numerical answers when evaluated on the others subset. Thus, we split the others subset into three small test sets, containing answer

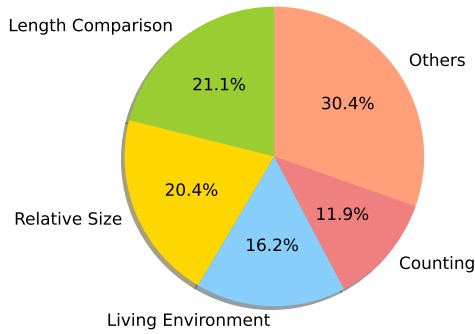

Figure 10: Detailed composition of the Others subset.

types of "yes/no", numbers, and other answers, respectively.

### B.3  More Qualitative Examples

We present additional qualitative examples in Table 6, which compare the predictions made by various models. The comparisons between OPT-7B and BLIP-2 demonstrate the effectiveness of incorporating visual information in enhancing the visual commonsense capabilities of LLMs. However, these leading models, including ChatGPT, also encounter difficulties in certain challenging cases, such as determining the color of a flamingo's beak tip. Besides, these examples highlight ChatGPT's tendency to prioritize selecting the most commonly associated property of an object as the answer, rather than considering the properties of the specific region in question. We hypothesize that this behavior may be attributed to the higher frequency of these common attributes co-occurring with the object in the pre-training text corpus.

| | | |
|---|---|---|
| **Related Images** |  | ... |
| **Question** | *What color is a goldfinch's tail?* | |
| **Answer** | OPT-7B: red ✗, BLIP-2: black ✓, ChatGPT: yellow ✗. | |
| **Related Images** |  | ... |
| **Question** | *What is the color of the flamingo's beak tip?* | |
| **Answer** | OPT-7B: green ✗, BLIP-2: red ✗, ChatGPT: pink ✗, (Ground Truth: black). | |
| **Related Images** |  | ... |
| **Question** | *What is the shape of the eyes of the jack-o'-lantern?* | |
| **Answer** | OPT-7B: cylinder ✗, BLIP-2: square ✗, ChatGPT: triangle ✓. | |
| **Related Images** |  | ... |
| **Question** | *What's the material of a china cabinet?* | |
| **Answer** | OPT-7B: cotton ✗, BLIP-2: wood ✓, ChatGPT: ceramic ✗. | |
| **Related Images** |  | ... |
| **Question** | *Does the koala have a tail?* | |
| **Answer** | OPT-7B: yes ✗, BLIP-2: no ✓, ChatGPT: yes ✗. | |

Table 6: More qualitative examples comparing different models on IMAGENETVC. Here we demonstrate comparisons with OPT-7B, BLIP-2, and ChatGPT.

| Prompts |
|---|
| [QUESTION] [ANSWER]. |
| [QUESTION] Answer: [ANSWER]. |
| [QUESTION] The answer is [ANSWER]. |
| Question: [QUESTION] Answer: [ANSWER]. |
| Question: [QUESTION] The answer is [ANSWER]. |

Table 7: The prompts we utilize for LLM and VaLM evaluation on IMAGENETVC.

## C Experimental Details

### C.1 Multiple Prompts

Table 7 shows all the intuitive prompt templates utilized to evaluate the LLMs and VaLMs in our main experiments. We do not tune the prompt for each subset in IMAGENETVC.

In our data quality experiments, we utilize the original prompts provided by Zhang et al. (2022a) for ViComTe evaluation while adopting the prompts in Table 7 for IMAGENETVC.

### C.2 Details of Human Assessment

In the human assessment process of ImageNetVC, annotators were presented with pairs of mutually exclusive random 32 instances from both ImageNetVC and other datasets (e.g., ViComTe) each time. In the whole process of comparison evaluation, we conducted multiple rounds of human assessment based on the dataset containing fewer test samples. Specifically, there are 556 comparison pairs used for evaluation between ImageNetVC and ViComTe, and 510 pairs used for evaluation between ImageNetVC and ChatGPT generated data.

Annotators were instructed to evaluate these instances based on the overall quality of the data, choosing the better side when considering three factors respectively: diversity, difficulty, and factuality. The final results were computed and demonstrated as percentages in Figure 4. For instance, the Diversity score depicted in Figure 4(a) signifies that in 86% of cases, the evaluators found ImageNetVC samples to either outperform or match those from

| Models | | #Param | COLOR | | SHAPE | | MATER. | | COMPO. | | OTHERS | | AVG |
|---|---|---|---|---|---|---|---|---|---|---|---|---|---|
| | | | 0-shot | 5-shot | 0-shot | 5-shot | 0-shot | 5-shot | 0-shot | 5-shot | 0-shot | 5-shot | |
| | Random* | - | 7.7 | 7.7 | 9.0 | 9.0 | 6.1 | 6.1 | 49.8 | 49.8 | 24.3 | 24.3 | 19.4 |
| OPT | OPT-125m | 125M | 16.3 | 7.6 | 18.7 | 12.4 | 25.2 | 24.8 | 51.0 | 50.7 | 38.4 | 39.4 | 28.5 |
| | OPT-350m | 350M | 19.9 | 23.1 | 18.4 | 25.8 | 27.7 | 26.0 | 50.4 | 51.4 | 38.2 | 42.3 | 32.3 |
| | OPT-1.3B | 1.3B | 26.5 | 27.1 | 32.7 | 42.6 | 37.3 | 41.3 | 50.3 | 51.7 | 44.2 | 53.0 | 40.7 |
| | OPT-2.7B | 2.7B | 25.8 | 33.4 | 39.9 | 49.7 | 40.2 | 58.7 | 51.3 | 54.5 | 48.1 | 55.6 | 45.8 |
| | OPT-6.7B | 6.7B | 31.1 | 37.1 | 38.6 | 51.2 | 46.1 | 61.4 | 53.4 | 61.5 | 51.9 | 52.9 | 48.5 |
| | OPT-30B | 30B | 31.7 | 41.1 | 40.5 | 59.2 | 49.8 | 70.4 | 59.2 | 66.3 | 54.4 | 59.7 | 53.2 |
| | OPT-66B | 66B | 34.1 | 44.0 | 40.6 | 57.1 | 47.3 | 68.4 | 56.1 | 64.3 | 55.5 | 60.8 | 52.8 |
| GPT | GPT-Neo-1.3B | 1.3B | 23.6 | 24.3 | 34.1 | 47.9 | 35.9 | 40.2 | 51.8 | 51.4 | 45.0 | 53.0 | 40.8 |
| | GPT-Neo-2.7B | 2.7B | 24.1 | 31.1 | 25.1 | 46.5 | 39.3 | 47.1 | 52.0 | 51.0 | 46.6 | 53.0 | 41.6 |
| | GPT-J-6B | 6B | 27.9 | 39.4 | 39.8 | 56.3 | 49.2 | 63.8 | 52.4 | 59.0 | 51.9 | 58.0 | 49.8 |
| Pythia | Pythia-160m | 160M | 18.6 | 23.0 | 24.7 | 34.8 | 24.3 | 32.0 | 51.0 | 50.1 | 39.8 | 42.2 | 34.1 |
| | Pythia-410m | 410M | 11.2 | 25.8 | 20.9 | 39.1 | 36.8 | 44.2 | 50.7 | 51.2 | 43.8 | 47.4 | 37.1 |
| | Pythia-1.4B | 1.4B | 18.8 | 29.8 | 36.5 | 41.5 | 39.0 | 55.6 | 50.9 | 50.5 | 47.3 | 52.4 | 42.2 |
| | Pythia-2.8B | 2.8B | 18.6 | 34.0 | 31.6 | 53.5 | 43.2 | 46.1 | 52.6 | 56.1 | 49.6 | 55.7 | 44.1 |
| | Pythia-6.9B | 6.9B | 24.7 | 38.4 | 30.7 | 52.6 | 42.2 | 60.8 | 54.2 | 55.7 | 51.9 | 56.5 | 46.8 |
| | Pythia-12B | 12B | 27.6 | 43.3 | 34.6 | 47.7 | 46.3 | 63.8 | 56.7 | 63.8 | 54.2 | 62.4 | 50.0 |
| Falcon | Falcon-7B | 7B | 36.9 | 50.3 | 35.8 | 60.4 | 55.8 | 69.8 | 59.3 | 75.5 | 54.9 | 58.4 | 55.7 |
| | Falcon-40B | 40B | 46.4 | 56.2 | 42.2 | 62.5 | 57.5 | 71.9 | 69.0 | 79.6 | 59.9 | 68.9 | 61.7 |
| LLaMA | LLaMA-7B | 7B | 41.0 | 52.8 | 34.9 | 54.4 | 50.9 | 69.9 | 64.3 | 72.4 | 57.0 | 58.9 | 55.7 |
| | LLaMA-13B | 13B | 41.9 | 55.7 | 40.8 | 65.9 | 52.0 | **72.1** | 59.0 | 79.2 | 58.5 | 64.2 | 58.9 |
| | LLaMA-30B | 30B | 46.6 | 60.5 | 41.5 | 52.7 | **58.2** | 72.0 | 68.5 | 82.8 | 59.4 | 70.3 | 61.3 |
| | LLaMA-65B | 65B | **50.8** | **64.0** | **46.7** | **70.4** | 57.5 | 71.8 | **71.3** | **84.0** | **61.1** | **73.0** | **65.1** |

Table 8: Evaluation results of large language models (LLMs) in IMAGENETVC. We report the mean Top-1 accuracy (%) over 5 different prompts. *: We report random results over 5 different runs for comparison. We show the best results in **boldface**.

ViComTe concerning diversity.

## C.3 Model Implementation

All the LLMs are implemented based on the Huggingface API[10]. For VaLMs, we utilize the official release of Z-LAVI[11], MAGMA[12], and BLIP-2[13] for evaluation. The CLIP model is adapted from the OpenAI's public source[14]. We utilize the vision-language pretrained checkpoints of VaLMs instead of task-specific finetuned models to evaluate their zero- and few-shot capabilities. The details of VaLMs, including the visual encoder architecture and pretraining data, are included in Appendix C.4. All hyperparameters of VaLMs are the same as that of their origin paper. We conduct our experiments on 6 NVIDIA A40 GPUs with 48GB memory.

## C.4 Model Details

We mainly conduct our evaluations with the following VaLMs in our experiments:

[10] https://huggingface.co
[11] https://github.com/YueYANG1996/Z-LaVI
[12] https://github.com/Aleph-Alpha/magma
[13] https://github.com/salesforce/LAVIS/tree/main/projects/blip2
[14] https://github.com/openai/CLIP

| Models | #Extra Params | Visual Encoder | #Pretrained Images |
|---|---|---|---|
| Z-LaVI | 150M | ViT-B/32 | - |
| MAGMA | 400M | RN50x16 | 25M |
| BLIP-2 | 1.1B | ViT-G/14 | 129M |

Table 9: Details of VaLMs evaluated on IMAGENETVC.

- **Z-LaVI** (Yang et al., 2022) introduces a zero-shot framework that ensembles the solutions of LLMs and CLIP (Radford et al., 2021) to handle plain language tasks. The extra visual inputs of CLIP are obtained with web search and synthesis methods.
- **MAGMA** (Eichenberg et al., 2022) adds additional adapter layers into frozen LLMs to augment them with visual capabilities. It also finetunes a visual encoder to transform images into visual prompts as prefixes.
- **BLIP-2** (Li et al., 2023a) introduces a Querying Transformer to extract visual features from the frozen image encoder. It then utilizes these features as visual prompts to augment frozen LLMs with visual information.

We show the details of VaLMs evaluated on IM-

| | Models | #Param. | COLOR | SHAPE | MATER. | COMPO. | OTHERS | AVG |
|---|---|---|---|---|---|---|---|---|
| | Random | - | 7.7 | 9.0 | 6.1 | 49.8 | 24.3 | 19.4 |
| VaLM (2.7B) | OPT-2.7B | 2.7B | 25.8 | 39.9 | 40.2 | 51.3 | 48.1 | 41.1 |
| | ⌐ w/ Z-LaVI | 2.9B | 37.3 ↑11.5 | 31.2 ↓8.7 | 46.6 ↑6.4 | 51.5 ↑0.2 | 46.4 ↓1.7 | 42.6 ↑1.5 |
| | ⌐ w/ BLIP-2 | 3.8B | 44.7 ↑18.9 | 40.3 ↑0.4 | 61.9 ↑21.7 | 54.0 ↑2.7 | 51.7 ↑3.6 | 50.5 ↑9.4 |
| VaLM (6B) | GPT-J-6B | 6B | 27.9 | 39.8 | 49.2 | 52.4 | 51.9 | 44.2 |
| | ⌐ w/ Z-LaVI | 6.2B | 38.2 ↑10.3 | 32.7 ↓7.1 | 51.9 ↑2.7 | 52.5 ↑0.1 | 48.6 ↓3.3 | 44.8 ↑0.6 |
| | ⌐ w/ MAGMA | 6.4B | 42.9 ↑15.0 | 40.3 ↑0.5 | 60.9 ↑11.7 | 54.6 ↑2.2 | 53.9 ↑2.0 | 50.5 ↑6.3 |
| VaLM (6.7B) | OPT-6.7B | 6.7B | 31.1 | 38.6 | 46.1 | 53.4 | 51.9 | 44.2 |
| | ⌐ w/ Z-LaVI | 6.9B | 38.8 ↑7.7 | 33.2 ↓5.4 | 49.5 ↑3.4 | 52.2 ↓1.2 | 49.6 ↓2.3 | 44.7 ↑0.5 |
| | ⌐ w/ BLIP-2 | 7.8B | 47.9 ↑16.8 | 39.1 ↑0.5 | 61.3 ↑15.2 | 58.4 ↑5.0 | 57.4 ↑5.5 | 52.8 ↑8.6 |

Table 10: Zero-shot probing results of visually-augmented language models (VaLMs) in IMAGENETVC. We report the mean accuracy (%) results in 5 different prompts. Numbers that are highlighted in orange represent the percentage of improvement and blue denotes the percentage of performance drop.

---

**Prompt 1**

Answer List: [CANDIDATES]
[QUESTION] Please select the most possible answer from the above list. Please answer in one word.

**Prompt 2**

Answer List: [CANDIDATES]
[QUESTION] Please only print the answer selected in the above list. Please answer in one word.

**Prompt 3**

[QUESTION] Please select the most possible answer from [CANDIDATES]. Please answer in one word.

Table 11: The prompts we utilize for ChatGPT evaluation on IMAGENETVC. "[CANDIDATES]" denotes the answer set of the evaluated subset in IMAGENETVC, as shown in Table 5.

AGENETVC in Table 9, including the extra model parameters (except the *frozen* LLM backbone), the architecture of the visual encoder, and the number of pretraining images. Since the VaLMs vary in implementation details (e.g., the visual encoder), we cannot make a direct (head-to-head) comparison between the VaLMs and leave it for future investigation.

## D  Details of Main Experimental Results

We provide detailed experimental numbers of our main results with LLMs and VaLMs in Table 8 and 10, respectively. For LLMs, we show the zero- and few-shot evaluation results of OPT, GPT, Pythia, Falcon, and LLaMA across various model scales. For VaLMs, we compare the performance of Z-LaVI, BLIP-2, and MAGMA with their *frozen* LLM backbones.

## E  Evaluation Details of Other Models

This section provides evaluation details of VQA finetuned multimodal models and RLHF models.

**Multimodal Models**  For multimodal models such as BLIP, we adopt the evaluation settings used in VQAv2 (Goyal et al., 2017). Specifically, for each QA pair and its corresponding image, we evaluate the model using open-ended generation and obtain the output answer. Based on the experimental settings outlined in Section 4.1.2, we provide the top-10 ranked images for each QA pair and determine the final answer by majority prediction.

**RLHF Models**  We evaluate ChatGPT with constrained prompts and automatically compute the top-1 accuracy. The prompts utilized are presented in Table 11.

## Short Instruction:

**Observe 20-50 images of a given category, find a vision feature of the given category, check if it conforms to most of the images and our common sense of life.**

**Then write a simple QA about this vision feature.**

The QA should contain one question and one correct answer.

**The vision feature can be:**

- **Coarse-grained**: color, shape, material, spotted/striped ... of the whole object.
- **Fine-grained**: color, shape, material ... of a certain part of the object.

Anyway, **the vision features are features that we can see through the images**. There are some features that can be referred to here.

---

**The Annotation pipeline is:**

- Look at the 20-50 images below, **find a common vision feature of the category**. For example, "The color of the Ibizan hound is yellow and white".
- Check if this feature conforms to **our common sense of life**.
- **Check if this feature is written.**
- One QA is created. For example, "What is the color of the Ibizan hound? yellow and white".

---

**The most cases we reject:**

1. **The QA has been written before.** Written QAs are shown below.

2. **You can not find the vision features which the QA tells about in the images. Or the QA is not talking about vision features but just common senses.**

   - **Common Sense**. For example, "Which country is rich in coconuts? Thailand" . It has nothing to do with visual features and can not be directly inferred from images.
   - **Usage**. "Can the camera be used to take pictures? yes", "Can the guitar be used to play music? yes", these questions are not talking about vision features.

3. **The QA is abstract feeling but not concrete features which can be directly seen from the images.** For example, "Is the rabbit cute? yes", or "Is the planetarium very large? yes", the absolute size can not be directly inferred from images, so the "big/small" question is not suitable.

4. **The QA does not conform to our common sense of life.** For example, "What is the color of the bottlecap? red". Not every bottlecap is red in our life.

5. **The QA is strongly related to images.** For example, "Are those apples all red in these images? yes". We hope to get the common vision attributes of the category, not of these images.

6. **The QA is not about the specific feature of the category.** For example, "Do the Chihuahua have legs? yes". We hope to get the specific feature of the category, like the body color of the Chihuahua (But in order to ensure the diversity of the data, we have accepted some such data before and will not accept it again).

**Click here to get Annotation Examples and full Reject Rules** (We will give **bonus** to workers who have a high accept rate).

### Category: Samoyed

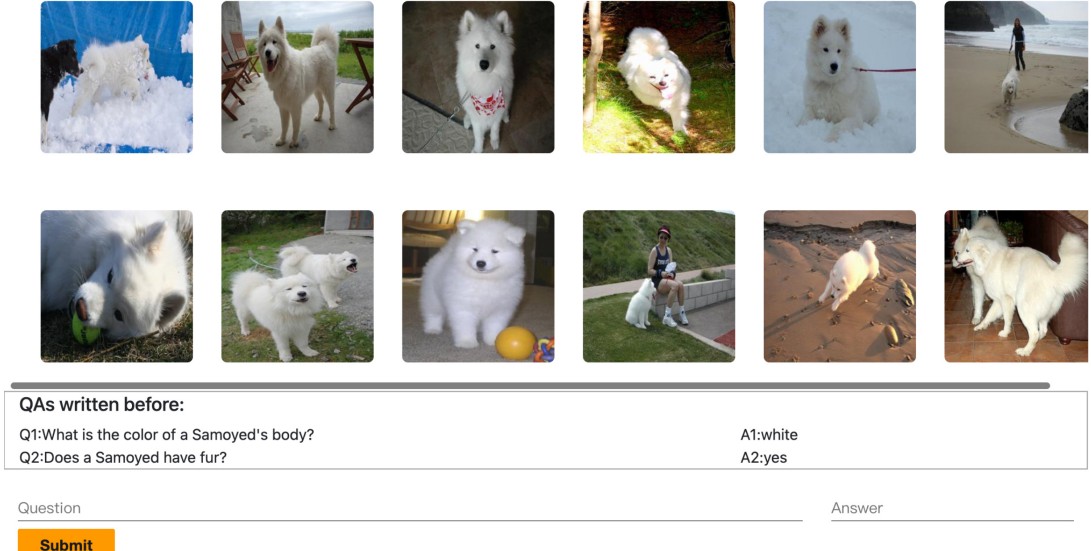

| QAs written before: | |
|---|---|
| Q1:What is the color of a Samoyed's body? | A1:white |
| Q2:Does a Samoyed have fur? | A2:yes |

| Question | Answer |
|---|---|

**Submit**

---

Figure 11: Screenshot of the IMAGENETVC annotation UI, featuring task instructions, the annotation pipeline, and the most common reasons for rejecting prompts during the annotation process. The interface displays 20-50 images of a given category, and the task instruction guides the annotator to identify a common vision feature and create a simple QA about it. The annotation pipeline includes checking the conformity to commonsense and avoiding pre-written QAs. Annotations are focused on visual features, not commonsense or non-visual attributes.