# OpenReview forum: "ImageNetVC: Zero- and Few-Shot Visual Commonsense Evaluation on 1000 ImageNet Categories"
_EMNLP/2023/Conference — EMNLP 2023 Findings_

### Official Review · Reviewer_MrcQ · 2023-08-06

**Soundness:** 4

**Excitement:**

4: Strong: This paper deepens the understanding of some phenomenon or lowers the barriers to an existing research direction.

**Missing References:**

NA

**Paper Topic And Main Contributions:**

This paper introduces a dataset with s 4,076 high quality QA pairs, encompassing 1,000 ImageNet categories across diverse domains such as color,shape, material, component, etc. For a given QA pair the authors provide multiple relevant images in some cases.The authors use manual human annotations so the data is claimed to be of higher quality. The authors test their dataset on popular uni(PLMs) and multimodal (VaLMs) models on visual commonsense reasoning tasks (visual grounding). The authors benchmark popular LLMs with various configurations (model scale, in-contect learning, image sources) on the visual commonsense reasoning task using their dataset. The authors also detail the data annotation process.

**Questions For The Authors:**

NA

**Reasons To Accept:**

The authors present a manually annotated dataset that they claim to have better alignment with QA and image pairs. They benchmark it with various popular Pretrained Language Models(PLMs) and visually augmented Language Models (VaLMs). High quality multimodal datasets and benchmarks on LLMs will be a good contribution to the research community. The authors provide clear benchmarking results and a commentary on how various LLMs behave (e.g., emergent properties) on their dataset. The paper is also well written and easy to follow.

**Reasons To Reject:**

The size of the dataset(4K) seems relatively low. It might also be better to have the model performance/questions on more dimensions than simply color, material, shape, component and pattern (e.g open domain VQA style questions.)

**Reproducibility:**

5: Could easily reproduce the results.

**Reviewer Confidence:**

4: Quite sure. I tried to check the important points carefully. It's unlikely, though conceivable, that I missed something that should affect my ratings.

---

> ### Author Rebuttal · Authors · 2023-08-27
>
> We sincerely thank Reviewer MrcQ for the positive feedback and we are grateful for the time you spent on our submission. We are also glad for the acknowledgment that our high-quality multimodal dataset is a good contribution to the research community, and our benchmarking results and paper writing are clear.
>
> Our response to your further comments is as follows:
>
> ***Q1: The size of the dataset(4K) seems relatively low.***
>
> R1: Thanks for your advice! We are committed to further expanding the dataset's size in our ongoing and future research endeavors. For instance, the dataset could be further expanded by prompting large language models with in-context demonstrations and then filtering with the image-text matching model. By doing so, we aim to enhance the dataset's coverage and its ability to serve as a comprehensive benchmark for zero- and few-shot visual commonsense evaluation. We are glad to explore this in the future and will highlight this in our revision.
>
> ***Q2: It might also be better to have the model performance/questions on more dimensions than simply color, material, shape, component and pattern (e.g open domain VQA style questions.)***
>
> R2: Thanks for your suggestion! The "Others" sub-task of ImageNetVC contains more diverse QA samples covering various topics such as length comparison (21%), relative size (20%), living environment (16%), counting (12%), etc. Visual commonsense knowledge assessment in open domain VQA style questions is a great point, and we are interested in constructing more challenging and large-scale datasets on visual commonsense.
>
> Thank you very much for the constructive comments, which really helped us further improve our work. We hope our answers have addressed your concerns. If you have any further questions, please feel free to let us know.

---

### Official Review · Reviewer_eD4G · 2023-08-06

**Typos Grammar Style And Presentation Improvements:** N/A
**Soundness:** 3

**Excitement:**

3: Ambivalent: It has merits (e.g., it reports state-of-the-art results, the idea is nice), but there are key weaknesses (e.g., it describes incremental work), and it can significantly benefit from another round of revision. However, I won't object to accepting it if my co-reviewers champion it.

**Missing References:**

N/A

**Paper Topic And Main Contributions:**

This paper introduced a new dataset covering different types of visual commonsense in 1000 classes from ImageNet. Multiple popular LLMs and VaLMs are benchmarked on this dataset. The authors show comprehensive experimental comparison and analysis, as well as several interesting observations.

**Questions For The Authors:**

See reasons to reject.

**Reasons To Accept:**

1. This paper introduced a new dataset about visual commonsense in 1000 categories of ImageNet. The data can be used to probe LLM and VaLM's understanding ability or knowledge of visual commonsense in a wide range of classes.
2. The paper benchmarked several prevailing and strong LLMs and VaLMs and present several insightful observations.

**Reasons To Reject:**

1. My main concern about the proposed dataset is that the type of knowledge it possesses is not very new from previous works, especially ViComTe. I went through examples of this paper's data and ViComTe's and didn't feel much difference. I.e., they all focus on different commonsense aspects of different objects, such as shape, material, and color, etc.  As for the differences listed in Tab1: (1). `Support VaLM`: I think other datasets can also be evaluated with VaLM. (2). `Region-based Question`: I think ViComTe also has region-based question? For example, in Fig1 of ViComTe, the shape or material of penguin all require locating at specific regions? (3). `Natural Language`: ViComTe uses a set of templates to generate data, which is also a kind of natural language. If the author is talking about whether the sentence is fluent and natural, I don't think it's a critical point because people can just rewrite template-based datasets using ChatGPT to make it natural and fluent.
2. It's not clear in the paper why category-level examination is necessary given that sample-level examination is gonna be conducted later. IMO, the sample-level examination is already enough.
3. In L222, I wonder what the average agreement rate is among all examples. Assume that 2/3 approval means 66.6% and 3/3 approval 100% for each sample.
4. This paper only evaluates VaLM models. I wonder why not evaluate VLM models such as MiniGPT4 and LLAVA?

**Reproducibility:**

4: Could mostly reproduce the results, but there may be some variation because of sample variance or minor variations in their interpretation of the protocol or method.

**Reviewer Confidence:**

4: Quite sure. I tried to check the important points carefully. It's unlikely, though conceivable, that I missed something that should affect my ratings.

---

> ### Author Rebuttal · Authors · 2023-08-27
>
> We sincerely thank Reviewer eD4G for your review and are also glad you think our dataset is valuable and our findings are insightful. Below we would like to give detailed responses to each of your comments.
>
> &nbsp;
>
> ***Q1: Comparison with previous works, especially ViComTe***
>
> R1: Yes, some previous works, such as ViComTe [1]*,* also worked on visual commonsense, which highlights the significance of this research problem. However, **one key difference between ImageNetVC and ViComTe is the way multimodal models are evaluated for visual commonsense**. ViComTe measures visual commonsense knowledge via textual frequency distributions over the corpora and evaluates multimodal models with **text-only inputs**, which has a misalignment with the pretraining data format (i.e. Text-Image pair) of multimodal models. Compared to ViComTe,  ImageNetVC proposes that **the visual commonsense knowledge in multimodal models demands relevant images as input to be suitably stimulated**, as we illustrate in **Table 3**. Equipped with diverse image sources (e.g., ImageNet, search images, synthetic images, and CLIP-ranked images), ImageNetVC provides **a fair VaLM evaluation with unified image sources** and facilitates future research on visual commonsense.
>
> In addition, previous datasets have not yet addressed the following three critical challenges, including **the quality, difficulty, and reliability** of the datasets. We would like to further clarify the distinct contributions of ImageNetVC that set it apart from prior datasets, which are threefold.
>
> 1. `Quality`: In contrast to ViComTe's automatically mined data from Visual Genome (VG) attributes, ImageNetVC directly leverages **human visual perception** to identify shared attributes in high-quality ImageNet images, **avoiding potential textual bias** and **providing data that has a closer alignment with human knowledge**. For example, we found low-quality mined samples in ViComTe, such as *(i, white)* and *(cat tail, black)*, which either contained non-visual objects or contained non-commonsense attributes. These two textual bias issues account for **7% and 34%** of all samples in ViComTe's color test set, respectively. Compared to ViComTe, the human annotation process (with quality verification) guarantees the high quality of our dataset.
> 2. `Difficulty`: Another advantage of ImageNetVC over prior work is that it incorporates **region-based visual commonsense** **questions**, which poses **greater challenges** due to the scarcity of such long-tail knowledge in the natural language corpora and the requirement of **the localization capabilities of PLMs**. Previous work, such as ViComTe, only focuses on the attributes of the whole object, neglecting the rich and meaningful visual commonsense knowledge from a fine-grained perspective. For example, the ViComTe color test set has samples of the form *(pine tree, green)*, *(kitchen drawer, white)* and *(taxi cab, yellow)*, which do not explicitly ask questions about specific regions.
> 3. `Reliability`: Furthermore, previous templated-based datasets, such as ViComTe, **exhibit severe prompt bias**, with substantial evaluation variance across different prompts (e.g., Accuracy varies from 2% to 61% as we illustrate in Figure 3). While ChatGPT can rewrite these template-based datasets, it may introduce **inconsistencies with the original data**. Figure 4(b) highlights potential risks associated with ChatGPT-generated results. We anticipate that large language models will enhance high-quality dataset development in the future. However, given the challenges of visual commonsense, manually constructed datasets are considered to be more reliable.
>
> As mentioned by *[Reviewer xCkP](https://openreview.net/forum?id=u14dVx4rMW&noteId=92wmzZ3JGo)*, our validation results in Section 3.5 of the paper further confirm the strengths of ImageNetVC over previous datasets (such as ViComTe) in terms of **factuality, difficulty, and diversity,** *demonstrating the need of human effort for the task of visual commonsense* *QAs*.
>
> &nbsp;
>
> ***Q2: It's not clear in the paper why category-level examination is necessary given that sample-level examination is gonna be conducted later. IMO, the sample-level examination is already enough.***
>
> R2: As agreed by *[Reviewer xCkP](https://openreview.net/forum?id=u14dVx4rMW&noteId=92wmzZ3JGo)*, we demonstrate a thorough annotation process involving cross-check examination of collected QA pairs and their categorization. Although sample-level examination can check each sample with high quality, it will consume more human resources and labeling costs. Therefore, we designed a **more efficient and cost-effective** stage - category-level examination, to filter out low-quality samples or those that do not meet our requirements.
>
> &nbsp;
>
> ***Q3: In L222, I wonder what the average agreement rate is among all examples. Assume that 2/3 approval means 66.6% and 3/3 approval 100% for each sample.***
>
> R3: In the stage of sample-level examination, **the average agreement rate is 78%**. This means that 78% of all samples are accepted during Stage 2. Among all these accepted samples, we observed that **72%** of them received unanimous approval from the examiners. For samples with disagreement among the reviewers (i.e., 2/3 approval), we authors conducted additional screening to ensure the data quality. Besides, we have implemented a hierarchical supervision process, as illustrated in Appendix A.3, where we authors monitored the examination quality during the annotation process. We will add more details of our annotation process to the final version.
>
> &nbsp;
>
> ***Q4: This paper only evaluates VaLM models. I wonder why not evaluate VLM models such as MiniGPT4 and LLAVA?***
>
> R4: ImageNetVC accommodates the evaluation of a wide range of models, including PLMs, VaLMs, and certainly VLMs and their instruction tuning backbones (e.g., Alpaca and Vicuna). However, we'd like to clarify that our main contribution is the constructed dataset and the research problem on the effectiveness of visual augmentation for PLMs. Therefore, we conduct our main experiments on PLMs & VaLMs and explore several co-founding factors that may affect the visual commonsense capability of models, such as model scale, in-context learning, and image sources. We have evaluated *LLaVA-13B* on ImageNetVC in the setting of open-ended generation. The results are illustrated below. We will also add the evaluation of *MiniGPT4 to the final version.*
>
> | Models     | Color | Shape | Material | Component | Others | Average |
> | ---------- | ----- | ----- | -------- | --------- | ------ | ------- |
> | Vicuna-13B | 57.3  | 64.2  | 71.9     | 77.8      | 67.3   | 67.7    |
> | LLaVA-13B  | 66.2  | 65.8  | 74.2     | 82.5      | 72.8   | 72.3    |
>
> The experimental results show that LLaVA achieves consistent improvements over its text-only Vicuna backbone, especially on the "Color" sub-task, indicating that visual instruction tuning could further boost the visual commonsense capability of models. All these additional experimental results will be included in our final manuscript.
>
> &nbsp;
>
> We sincerely hope that the above clarifications will resolve your concerns. Feel free to leave a comment if you have questions for further discussion.
>
> **Reference**
>
> [1] Zhang et al. Visual Commonsense in Pretrained Unimodal and Multimodal Models. In NAACL 2022.

---

### Official Review · Reviewer_xCkP · 2023-08-06

**Soundness:** 4

**Excitement:**

4: Strong: This paper deepens the understanding of some phenomenon or lowers the barriers to an existing research direction.

**Paper Topic And Main Contributions:**

The paper proposes a new dataset named ImageNetVC, designed for zero- and few-shot visual commonsense evaluation of Pretrained Language models (PLM) such as OPT, LLAMA, Falcon, and ChatGPT and Vision and Language models (VaLM) across the ImageNet categories. The evaluation is done as a QA set up with human annotation of QA pairs focused on color, shape, and material of classes.
To validate their dataset, the authors run another stage of cross-examine validation with human workers, and studies of human preference to show that their dataset is interesting and challenging than another visual commonsense dataset (ViComTe), and one generated by language models such as ChatGPT.  Extensive experiments are run to measure the factors that influence the performance of PLMs and VaLMs in their visual commonsense understanding, such as in-context-learning, image sources, and model size.
Human evaluation shows that their human annotation scheme provides much higher-quality than template and language model generated datasets, and the gap between machine-level (75.8%) and human-level (93.5%) of visual commonsense suggests that there is a good room of improvement to investigate further in this dataset.  Overall, the authors run thorough experiments to analyze possible factors of model performance and human evaluations to validate their dataset annotations.

**Questions For The Authors:**

Questions are mostly re-phrases of weaknesses:
A. How do we agree that answers are valid for all classes? For example, the color of "husky" is "black" and "white", but some of them could have the color "brown". It would be good to get more qualitative examples of the collected data.
B. What are some qualitative examples when LM and VaLM fail on? What is the missing gap between machine vs human performance of visual commonsense that should be worked on?


**Reasons To Accept:**

Overall, the paper is well-written, running extensive studies in validating their dataset annotations and possible baselines related to their task with in-depth analysis. Here are its strengths:
- demonstrates thorough annotation steps involving cross-check examination of collected QA pairs and their categorization.
- runs another set of human evaluation to claim the value of their ImagenetVC dataset; e.g. Figure 4 shows that humans found the ImageNetVC dataset to be "better" in terms of diversity, difficulty, and factuality than previous similar dataset  (ViComTe) and machine generated (ChatGPT). This proves the need of human effort for the task of visual commonsense QAs, which is seldom explored in similar dataset papers.
- explores performance of various, SoTA pre-trained text-only (e.g. LLAMA, ChatGPT) and vision-language models (BLIP-2), and runs analysis of which categories these models typically excel on. The results show that the task cannot be solved by pre-trained language models and that the image modality is still required to do achieve human-level performance.
- shows insightful findings in their results such as:
   - in-context learning (ICL) can lead to reduced variance across different prompts and improved calibration of language models (Figure 8), and improve the VaLM's visual commonsense performance.
  - an ensemble of both generated and retrieved images as input could boost the performance of visual commonsense understanding.
  - Scaling law of PLMs (Figure 7).
- Shows that human performance is 93.5% v.s. the best model (in context learning with ChatGPT) is 75.8%, meaning that there is still a lot of room to improve on and this is a well-designed task.

**Reasons To Reject:**

- Using visual attributes such as color, material and shape seem to be far-fetched to be called as measuring "visual commonsense". They seem to be mostly "factual knowledge" that require perception and recognition of objects, rather than requiring reasoning.
- Lack of qualitative examples other than the Samoyed example.
- Failure cases to understand what is the gap between current PLMs & VaLMs v.s. humans in visual commonsense.

**Reproducibility:**

4: Could mostly reproduce the results, but there may be some variation because of sample variance or minor variations in their interpretation of the protocol or method.

**Reviewer Confidence:**

3: Pretty sure, but there's a chance I missed something. Although I have a good feel for this area in general, I did not carefully check the paper's details, e.g., the math, experimental design, or novelty.

**Typos Grammar Style And Presentation Improvements:**

- Table and Radar plot showing LMs and VaLMs at the same time would helpful to see what are the categories that unimodal vs multimodal models excel on.

---

> ### Author Rebuttal · Authors · 2023-08-27
>
> We deeply appreciate your effort in reviewing our paper and your acknowledgment of our paper’s contribution. We are very glad that you like our task design, in-depth analysis, and insightful findings.
>
> Below we give point-to-point responses to each of your comments.
>
> ***Q1: Using visual attributes such as color, material and shape seem to be far-fetched to be called as measuring "visual commonsense". They seem to be mostly "factual knowledge" that requires perception and recognition of objects, rather than requiring reasoning.***
>
> R1: We'd like to clarify that our work focuses on the atomic visual commonsense knowledge. Such fundamental commonsense serves as the basis for more intricate reasoning processes. Our work can be likened to a visual counterpart of ConceptNet [1], as we aim to establish a foundational understanding of the visual commonsense of PLMs and VaLMs.
>
> More concretely, we follow ViComTe's definition [2] of "visual commonsense", i.e., knowledge about generic visual concepts such as "knobs are usually round", which allows for a straightforward assessment of the visual capabilities of both PLMs and VaLMs. We agree that more sophisticated benchmarks for visual commonsense reasoning (e.g., [3]) are also crucial for PLMs, but it is beyond the scope of our paper. We leave this insightful study for future investigation.
>
> ***Q2: What are some qualitative examples of when LM and VaLM fail? What is the missing gap between machine vs human performance of visual commonsense that should be worked on?***
>
> R2: We appreciate your inquiry for more qualitative examples. In the final version of the manuscript, we will incorporate more qualitative examples and case studies that highlight instances where PLMs and VaLMs struggle, to better illustrate the limitations and gaps.
>
> ***Q3: How do we agree that answers are valid for all classes? For example, the color of "husky" is "black" and "white", but some of them could have the color "brown". It would be good to get more qualitative examples of the collected data.***
>
> R3: We appreciate your inquiry for more qualitative examples of the collected data. We will illustrate more examples in the final manuscript. As demonstrated in Appendix A of our paper, during the annotation process, we asked the annotators to write visual features that match the majority of the given images and align with their commonsense of life. For example, "The shape of the dorsal fin of the tiger shark is triangular". To ensure the reliability and clarity of our dataset, questions with multiple possible (and confusing) answers (such as *the color of a cat's tail*) do not meet our requirements and will not be accepted during the examination.
>
> We greatly appreciate the valuable feedback you provided, which significantly contributed to enhancing our work. We hope that the above clarifications will resolve your concerns. If you have any further questions, please feel free to let us know.
>
> **Reference**
>
> [1] Robyn et al. Conceptnet 5.5: An open multilingual graph of general knowledge. In AAAI 2017.
>
> [2] Zhang et al. Visual Commonsense in Pretrained Unimodal and Multimodal Models. In NAACL 2022.
>
> [3] Rowan et al. From Recognition to Cognition: Visual Commonsense Reasoning. In CVPR 2019.

---

### Meta-Review · Area_Chair_a5qJ · 2023-09-14

**Recommendation:** 3

**Metareview:**

Based on the reviews and my perusal of the manuscript, the work clearly states its goal---manually curating a benchmark dataset for visual commonsense QA based on ImageNet. The authors have also prepared quite a thorough comparison with existing ViComTe dataset in terms of *diversity*, *difficulty*, and *factuality* (correctness). They also evaluate LLMs and VaLMs (Vision-augmented LMs) on the curated ImageNetVC dataset under zero- and few-shot settings. The authors did not perform evaluation on instruction-tuned VLMs, as pointed out by  eD4G. They subsequently provided the results in the rebuttal. Overall, ImageNetVC appears more difficult, factual, and diverse. They also provide many interesting observations and analyses w.r.t. various aspects of the image, the effect of in-context learning, emergence of visual commonsense with model scale, model calibration for ICL, lack of prompt bias of ImageNetVC samples over ViComTe, etc.

# Pros
1. Overall, the authors seem to have tried their best to leave few questions unanswered in the manuscript.
2. The overall goal and the aim and conclusions of the experiments seem mostly clear to the reviewers, especially after the rebuttal.
3. This dataset could be significant in evaluating fundamental visual knowledge of LLM/VLMs, if not for more complex visual commonsense reasoning. According to the presented empirical human and automatic evaluation, ImageNetVC seems to be more challenging, yet factual and diverse dataset.

# Cons (The authors may want to consider some of these points in the future iterations)
1. The end product, the methods, and the evaluation are not particularly original. There is already ViComTe dataset with similar purpose. The zero- and few-shot evaluations of LLM and VLMs across various aspects is quite standard in NLP at this point. As pointed out by eD4G, the current manuscript claim this as novelty.
2. It is unclear how A/B human evaluation was performed in fig. 4 (by category?). How do you get the notion of parallel instances across the datasets? Also, diversity is a global aspect of the dataset, unlike factuality and difficulty that can be evaluated per QA pair. More details are required here.
3. The authors claim that PLMs may emerge to possess zero- and few-shot visual commonsense from 1.3B size onwards. However, such claim is made based on a sample size of only two. The authors also do not mention why it could be. Do they share any part of the pre-training dataset? That could lead to them having similar performance curves for visual commonsense. As far as I know, both at least share PILE as the pre-training dataset. More analysis is required.
4. More qualitative examples should be added as mentioned by reviewer xCkP.
5. The rationale and difference between the category- and instance-level filtering is not quite clear, even after rebuttal. Are you not looking at the all instances during category-level cross-checking? If so, why the instance-level cross-check once again? Can you elaborate on the distribution bias? Couldn't you have just done sample-level cross-check, followed by more annotation for categories with significantly fewer samples?

---

### Decision · Program_Chairs · 2023-10-07

**Decision:**

Accept-Findings

**Comment:**

Based on the reviews and my perusal of the manuscript, the work clearly states its goal---manually curating a benchmark dataset for visual commonsense QA based on ImageNet. The authors have also prepared quite a thorough comparison with existing ViComTe dataset in terms of *diversity*, *difficulty*, and *factuality* (correctness). They also evaluate LLMs and VaLMs (Vision-augmented LMs) on the curated ImageNetVC dataset under zero- and few-shot settings. The authors did not perform evaluation on instruction-tuned VLMs, as pointed out by  eD4G. They subsequently provided the results in the rebuttal. Overall, ImageNetVC appears more difficult, factual, and diverse. They also provide many interesting observations and analyses w.r.t. various aspects of the image, the effect of in-context learning, emergence of visual commonsense with model scale, model calibration for ICL, lack of prompt bias of ImageNetVC samples over ViComTe, etc.

# Pros
1. Overall, the authors seem to have tried their best to leave few questions unanswered in the manuscript.
2. The overall goal and the aim and conclusions of the experiments seem mostly clear to the reviewers, especially after the rebuttal.
3. This dataset could be significant in evaluating fundamental visual knowledge of LLM/VLMs, if not for more complex visual commonsense reasoning. According to the presented empirical human and automatic evaluation, ImageNetVC seems to be more challenging, yet factual and diverse dataset.

# Cons (The authors may want to consider some of these points in the future iterations)
1. The end product, the methods, and the evaluation are not particularly original. There is already ViComTe dataset with similar purpose. The zero- and few-shot evaluations of LLM and VLMs across various aspects is quite standard in NLP at this point. As pointed out by eD4G, the current manuscript claim this as novelty.
2. It is unclear how A/B human evaluation was performed in fig. 4 (by category?). How do you get the notion of parallel instances across the datasets? Also, diversity is a global aspect of the dataset, unlike factuality and difficulty that can be evaluated per QA pair. More details are required here.
3. The authors claim that PLMs may emerge to possess zero- and few-shot visual commonsense from 1.3B size onwards. However, such claim is made based on a sample size of only two. The authors also do not mention why it could be. Do they share any part of the pre-training dataset? That could lead to them having similar performance curves for visual commonsense. As far as I know, both at least share PILE as the pre-training dataset. More analysis is required.
4. More qualitative examples should be added as mentioned by reviewer xCkP.
5. The rationale and difference between the category- and instance-level filtering is not quite clear, even after rebuttal. Are you not looking at the all instances during category-level cross-checking? If so, why the instance-level cross-check once again? Can you elaborate on the distribution bias? Couldn't you have just done sample-level cross-check, followed by more annotation for categories with significantly fewer samples?